# Gender effect in survival after out-of-hospital cardiac arrest: A nationwide, population-based, case-control propensity score matched study based Korean national cardiac arrest registry

**Han Zo Choi** [1], **Hansol Chang** [2,3], **Seok Hoon Ko** [4], **Myung Chun Kim** [1] *

1 Department of Emergency Medicine, Kyung Hee University Hospital at Gangdong, Seoul, South Korea,
2 Department of Emergency Medicine, Samsung Medical Center, Sungkyunkwan University School of Medicine, Seoul, South Korea, 3 Department of Digital Health, Samsung Advanced Institute for Health Science & Technology (SAIHST), Sungkyunkwan University, Seoul, South Korea, 4 Department of Emergency Medicine, Kyung Hee University Medical Center, Seoul, South Korea

* edkmc@khnmc.or.kr

**Data Availability Statement:** All OHCA data files are available from the Korea Disease Control and

## Abstract

### Objective

This study aimed to describe the relationship between sex and survival of patients with out-of-hospital cardiac arrest (OHCA) and further investigate the potential impact of female reproductive hormones on survival outcomes, by stratifying the patients into two age groups.

### Methods

This retrospective, national population-based observational, case-control study, included Korean OHCA data from January 1, 2009, to December 31, 2016. We used multiple logistic regression with propensity score-matched data. The primary outcome was survival-to-discharge.

### Results

Of the 94,160 patients with OHCA included, 34.2% were women. Before propensity score matching (PSM), the survival-to-discharge rate was 5.2% for females and 9.1% for males, in the entire group (OR 0.556, 95% CI [–0.526–0.588], $P<0.001$). In the reproductive age group (age 18–44 years), the survival-to-discharge rate was 14% for females and 15.6% for males (OR 0.879, 95% CI [0.765–1.012], $P = 0,072$) and in the post-menopause age group (age $\geq$ 55 years), the survival-to-discharge rate was 4.1% for females and 7% for males (OR 0.562, 95% CI [0.524–0.603], $P<0.001$). After PSM (28,577 patients of each sex), the survival-to-discharge rate was 5.4% for females and 5.4% for males (OR, 1.009 [0.938–1.085], $P = 0.810$). In the reproductive age group, the survival-to-discharge rate was 14.5% for females and 11.5% for males (OR 1.306, 95% CI [1.079–1.580], $P = 0.006$) and in the

Prevention Agency database. http://www.kdca.go.kr/contents.es?mid=a20303010403.

**Funding:** The authors received no specific funding for this work.

**Competing interests:** The authors have declared that no competing interests exist.

post-menopause age group, the survival-to-discharge rate was 4.2% for females and 4.6% for males (OR 0.904, 95% CI [0.828–0.986], $P$ = 0.022). After adjustment for confounders, women of reproductive age were more likely to survive at hospital discharge. However, there was no statistically significant difference in neurological outcome (OR 1.238, 95% CI [0.979–1.566], $P$ = 0.074).

## Conclusions

Females of reproductive age had a better chance of survival when matched for confounding factors. Further studies using sex hormones are needed to improve the survival rate of patients with OHCA.

## Introduction

Out-of-hospital cardiac arrest (OHCA) is a major cause of mortality worldwide. Approximately 350,000 Americans suffer OHCAs annually, with the overall survival rate being 12% in 2016 [1]; further, there have been approximately 29,800 OHCAs in Korea in 2016, with an overall survival rate of 7.6% [2].

Several investigators believe that there are sex-specific differences in survival outcomes in patients with OHCA. Reports regarding sex-based differences in outcomes are conflicting, with some studies showing comparable survival between both sexes [3, 4], and other studies showing comparable but better survival in females of reproductive age [5–8]. Certain studies have suggested that this difference is due to the protective effects of endogenous estrogen in females [9, 10]. Animal studies have shown that estrogen administration may improve cardiac arrest outcomes [11, 12]. Contrarily, some studies have reported that the OHCA survival rate among females in the reproductive age, was similar to that in males [13, 14]. Further research is required to explain these conflicting results.

A previous study [9] has reported differences in the baseline characteristics of women and men with OHCA. Compared with women, men were younger, were more likely to have witnessed OHCA, and had a higher frequency of bystander cardiopulmonary resuscitation (CPR) and initial shockable rhythm. In the present study, the basic differences in baseline characteristics between men and women were eliminated by propensity score matching, in order to determine the role of sex hormones in survival after OHCA. Furthermore, instead of sex hormone levels, the groups were divided into reproductive and non-reproductive age groups [9]. To date, no study has been conducted using a nationwide propensity score-matched data, to evaluate sex-specific survival after OHCA.

Our objective was to use a nationwide Korean population-based research database of OHCA, to describe the relationship between sex and survival in patients with OHCA, and further stratify them into two age groups [6, 7] for evaluation of the potential impact of female reproductive hormones on survival outcomes, using the propensity score matching (PSM) method to control covariates that produce selection bias.

## Materials and methods

### Study design and setting

This nationwide population-based observational study included 94,160 adult patients aged >17 years. In South Korea, 51.8 million people (2015 census) reside in an area of approximately 100,000 km$^2$. The Korean emergency medical services (EMS) are a single-tier,

government-provided system headed by the National Emergency Management Agency, which provides advanced cardiac life support (ACLS) and basic life support ambulance services throughout the 16 provincial headquarters.

The ambulance crew is trained to administer CPR and apply automatic external defibrillation at the scene and during transport; in limited cases, the ACLS-trained crew can also perform ACLS on-site under the direction of a physician. This includes administering intravenous fluids, inserting an endotracheal tube, and administering certain medications such as epinephrine and atropine, with the directions of a physician. Emergency medical technicians (EMTs) are not permitted to pronounce death, and they cannot stop CPR in the field unless return of spontaneous circulation (ROSC) occurs. Therefore, all OHCA patients who are treated by EMS personnel, are transported to the hospital emergency department (ED). There are situations in which CPR for cardiac arrest is not initiated at the scene or during ambulance transport. If the patient meets the eligibility criteria for withdrawal of resuscitation, then the EMTs are not permitted to start CPR. The criteria include prolonged arrest, decapitation or decomposition of the body, onset of rigor mortis, and livor mortis. The ED physicians can decide whether to continue or discontinue CPR, even in the setting of the EMS crew performing CPR during transport.

This study was approved by the institutional review boards of the participating institutions (IRB number: 2021-05-005), and the need for informed consent was waved.

## Data collection and process

This study used a nationwide, population-based, EMS-assessed OHCA database covering the entire country [15, 16]. The database was built from ambulance-run sheets. Review of the run sheets was followed by a review of the subsequent hospitalization records for each patient. Database construction began in 2006, and the database is maintained at present (2020) with support from the Korean Centers for Disease Control and Prevention and the National Emergency Management Agency.

If apply for the use of raw data on the Korea Centers for Disease Control and Prevention site, OHCA data can be used with the consent of the official. The database comprises geographical and sociodemographic data, location of the cardiac arrest, elapsed time variables associated with resuscitation efforts (response time and transport time), content of treatments, and destination hospitals. A Korean Centers for Disease Control and Prevention expert who is trained in medical record review is responsible for reviewing the hospital records. The review form has been modeled on the Utstein-style report form and was customized for this study setting [3].

## Selection and description of participants

All patients included in this study were adults with OHCA, >17 years, and with presumed cardiac etiology. OHCA cases from January 1, 2009, to December 31, 2016, were reviewed. Patients having arrests due to trauma, drowning, asphyxia, hanging, or other obvious non-cardiac causes were excluded. Exclusion criteria were based on the definition of the Utstein taxonomy. The characteristics included in the dataset were as follows: patient's sex and age, place of arrest (home vs. outside of home), presence of bystander witnesses, maneuver of bystander CPR, initially identified cardiac rhythm (shockable vs. non-shockable), use of prehospital defibrillation, presence of ROSC, result of ED treatment, result of hospital treatment, and neurological status at discharge. Neurological status was defined using the Cerebral Performance Category (CPC) scale scores (1, good cerebral performance; 2, moderate cerebral disability; 3, severe cerebral disability; 4, coma or vegetative state; and 5, death) [17].

## Outcome measure

Survival-to-hospital discharge (discharged alive/remained in-hospital 30 days post-arrest) was the primary outcome. Secondary outcomes were ROSC at the scene or in the ED, survival-to-hospital admission, and survival with good neurological status with an overall post-arrest CPC scale score of 1 or 2.

## Statistical analysis

Continuous variables are presented as mean and standard deviation. Categorical variables are presented as numbers and percentages. Patients were divided into two groups (male and female). To compare the two groups, Student's t-test was used for continuous variables and the chi-square test was used for categorical variables. To eliminate the effect of confounding variables that influence outcome variables, when analyzing basic characteristics, the PSM method was used to collect data in both groups. Women patients were matched 1:1 with male patients according to the propensity score, using exact matching. To assess bias reduction in the PSM method, absolute standardized differences were calculated, with a value of >20%, indicating a significant imbalance in the baseline covariate. Using matched data, differences between male and female outcome variables were analyzed again. If significant variables were found on comparing the matched data of both sexes, then multivariate logistic regression analysis was performed with these significant variables. In Korea, the mean age for natural menopause in women is approximately 49 years [18]. Although there are other age group definitions defined for reproductive age, studies have consistently used the age range of 18–44 years as measurement of reproductive age [19]. We also analyzed two subgroups, aged 18–44 years and ≥55 years (excluding the peri-menopausal group aged 45–54 years), to assess the association between estrogen exposure and survival [7, 13, 20].

All statistical analyses were performed using R software (version 3.6.2 0 (R Foundation for Statistical Computing, Vienna, Austria). P-values were based on a two-sided significance level of 0.05.

# Results

## Characteristics of study subjects

Total 214,954 patients of OHCA were identified in this study from January 1, 2009, to December 31, 2016. Patients with arrests due to trauma or of unknown origin (n = 57,153), no resuscitation attempted by EMS (n = 47,752), arrests of non-cardiac etiology (n = 10,181), under 18 years of age (n = 5,308) and having no age records (n = 400) were excluded from the study. Finally, 94,160 patients were included. There were 8,465 patients between 18 and 44 years and 72,119 patients in the age group of 55 years and above (Fig 1).

Patient matching was achieved in 61.1% (57,514 of 94,160) of all patients, 44.9% (3,798 of 8,465) in those aged 18–44 years, and 67.6% (48,756 of 72,119) in those aged over 55 years (Fig 2).

## Main results

This study found that women (N = 32,345) had much less OHCA than men (N = 61,915), were older than men and their OHCA occurred more at home than outside of home. They were less likely, to experience a witnessed arrest, have an initial shockable rhythm, receive bystander CPR, and prehospital defibrillation (Table 1).

In the subgroup aged 18–44 years, women were younger, more likely to experience OHCA at home, less likely to be witnessed, have an initial shockable rhythm, and receive bystander

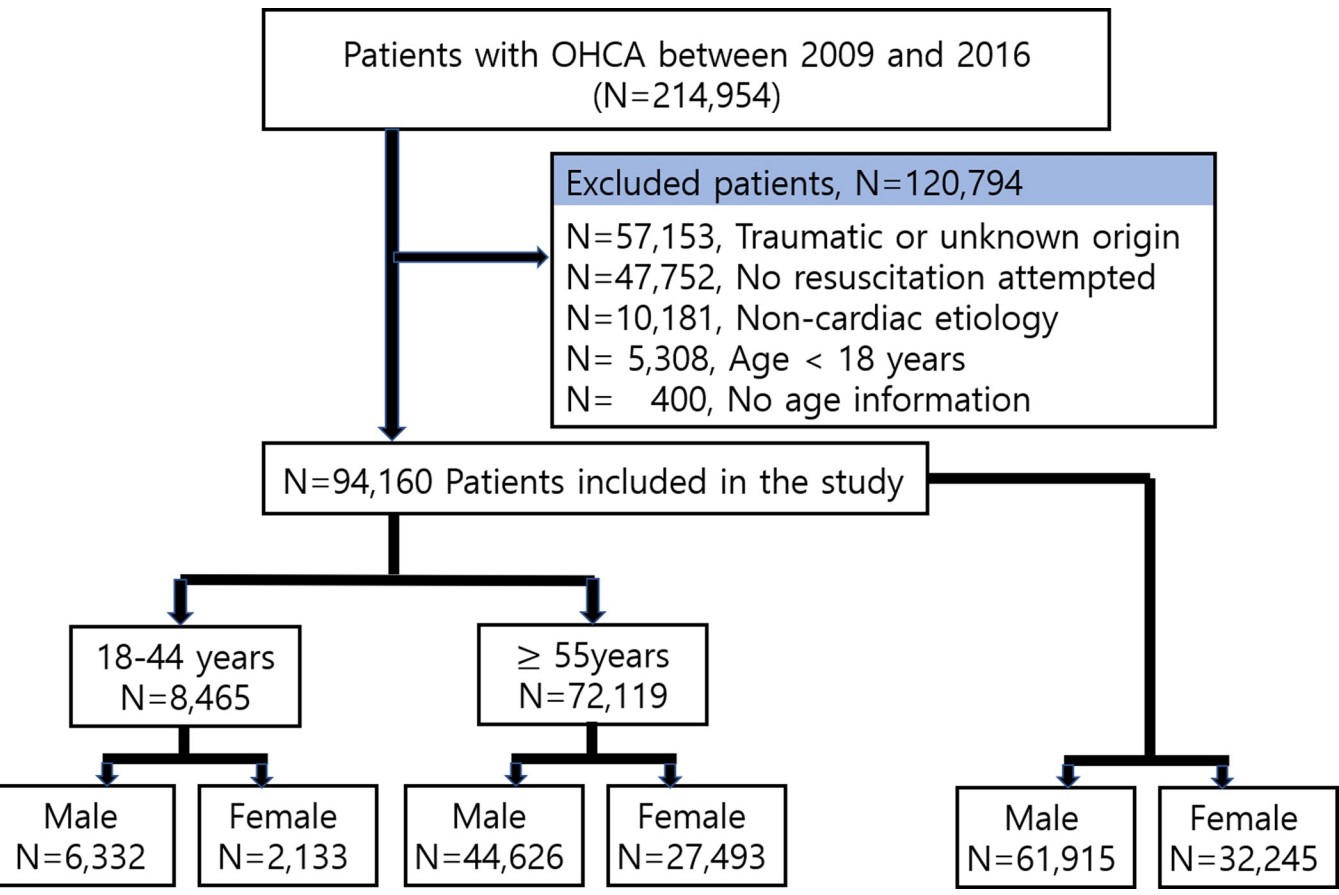

**Fig 1. Inclusion and exclusion flow chart.** We had excluded patient step by step from top to above.

CPR and prehospital defibrillation, than men. In the age group of 55 years and above, women were older, more likely to experience OHCA at home, less likely to be witnessed, have an initial shockable rhythm, and receive bystander CPR and prehospital defibrillation, than men (Table 2).

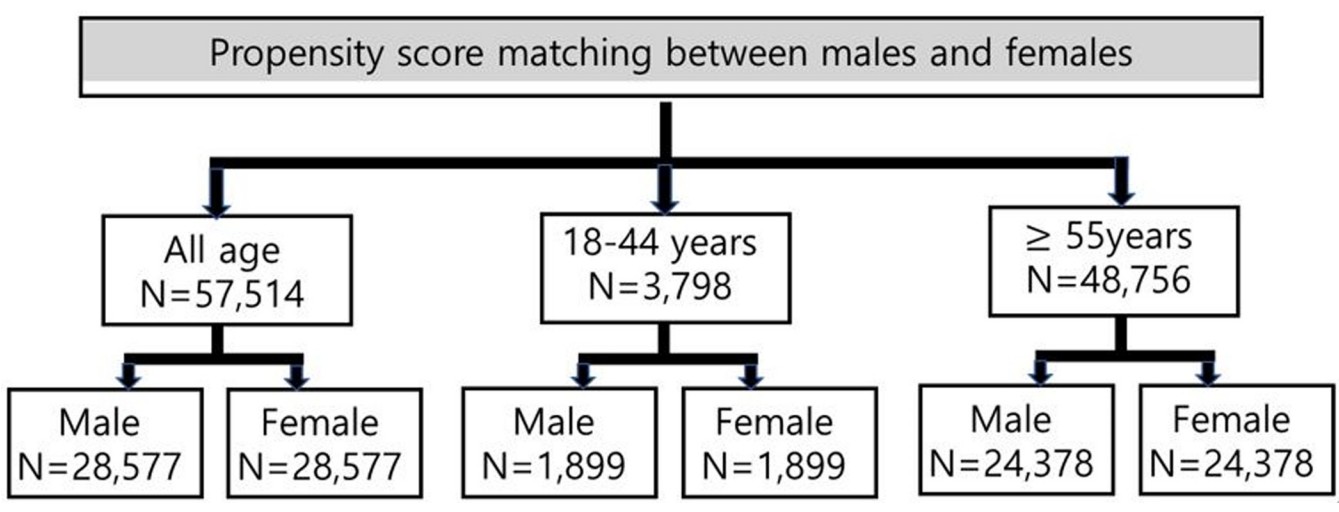

**Fig 2. Inclusion after propensity score matching.**

**Table 1. Characteristics of patients with out-of-hospital arrest by sex (total number = 94,160).**

| Variables | Male (N = 61,915) | Female (N = 32,245) | *P*-value | Standardized difference |
|---|---|---|---|---|
| Age (mean ± SD) | 63.6 ± 14.8 | 71.7 ± 15.2 | <0.001 | 3.63664 |
| Place of arrest | | | <0.001 | −0.12886 |
| Home residence | 32,749 (60.8) | 19,142 (67) | | |
| Outside of home | 21,108 (39.2) | 9,435 (33) | | |
| Witnessed arrest | 32,825 (53) | 16,278 (50.5) | <0.001 | −0.05072 |
| Bystander CPR | 9,543 (15.4) | 4,340 (13.5) | <0.001 | −0.05561 |
| Initial shockable rhythm | 7,811 (12.6) | 1,917 (5.9) | <0.001 | −0.23142 |
| Prehospital defibrillation | 11,118 (18) | 3,032 (9.4) | <0.001 | −0.25087 |
| ROSC at scene or ED | 4,719 (7.6) | 1,490 (4.6) | <0.001 | |
| Survival-to-admission | 15,631 (25.2) | 6,733 (20.9) | <0.001 | |
| Survival-to-discharge | 5,609 (9.1) | 1,692 (5.2) | <0.001 | |
| CPC 1 or 2 | 3,259 (5.3) | 784 (2.4) | <0.001 | |

SD, standard deviation; CPR, cardiopulmonary resuscitation; ROSC, return of spontaneous circulation; ED, emergency department; CPC, cerebral performance category

After matching male and female patients, we were able to analyze 28,577 pairs of patients over 18 years of age, 1,899 pairs of patients aged between 18 and 44 years, and 24,378 pairs of patients over 55 years of age. After matching data for the group of patients over 18 years of age, a comparison between the sexes showed differences in the age, place of arrest, and initial shockable rhythm variables. The outcome variables showed no difference between men and women (Table 3).

The matched data for the 18–44 age group of patients, showed no statistically significant variables, on comparison between men and women. The outcome variables showed that

**Table 2. Characteristics of patients with out-of-hospital arrest by age group and sex.**

| Variables | 18–44 years (Total number = 8,465) | | | | ≥ 55 years (Total number = 72,119) | | | |
|---|---|---|---|---|---|---|---|---|
| | Male (N = 6,332) | Female (N = 2,133) | *P*-value | Standardized difference | Male (N = 44,626) | Female (N = 27,493) | *P*-value | Standardized difference |
| Age (mean ± SD) | 36.3 ± 6.9 | 35.2 ± 7.4 | <0.001 | −2.10271 | 70.8 ± 9.5 | 76.6 ± 9.6 | <0.001 | 6.38088 |
| Place of arrest | | | <0.001 | −0.24259 | | | <0.001 | −0.08971 |
| Home residence | 3,175 (58) | 1,322 (69.6) | | | 24,254 (62.2) | 16,211 (66.5) | | |
| Outside of home | 2,295 (42) | 577 (30.4) | | | 14,737 (37.8) | 8,167 (33.5) | | |
| Witnessed arrest | 3,540 (55.9) | 1,097 (51.4) | <0.001 | −0.08986 | 23,274 (52.2) | 13,486 (50.4) | <0.001 | −0.03584 |
| Bystander CPR | 1,199 (18.9) | 362 (17) | 0.043 | −0.05119 | 6,527 (14.6) | 3,598 (13.1) | <0.001 | −0.04455 |
| Initial shockable rhythm | 1,299 (20.5) | 247 (11.6) | <0.001 | −0.24525 | 4,526 (10.1) | 1,396 (5.1) | <0.001 | −0.19187 |
| Prehospital defibrillation | 1,721 (27.2) | 360 (16.9) | <0.001 | −0.25051 | 6,642 (14.9) | 2,256 (8.2) | <0.001 | −0.21012 |
| ROSC at scene or ED | 787 (12.4) | 248 (11.6) | 0.328 | | 2,694 (6) | 1,001 (3.6) | <0.001 | |
| Survival-to-admission | 2,012 (31.8) | 795 (37.3) | <0.001 | | 10,136 (22.7) | 5,031 (18.3) | <0.001 | |
| Survival-to-discharge | 987 (15.6) | 298 (14) | 0.072 | | 3,131 (7) | 1,119 (4.1) | <0.001 | |
| CPC 1 or 2 | 681 (10.8) | 185 (8.7) | 0.006 | | 1,572 (3.5) | 430 (1.6) | <0.001 | |

SD, standard deviation; CPR, cardiopulmonary resuscitation; ROSC, return of spontaneous circulation; ED, emergency department; CPC, cerebral performance category

**Table 3. Characteristics of patients with out-of-hospital arrest by sex after propensity score matching (total number = 57,514).**

| Variables | Male (N = 28,577) | Female (N = 28,577) | *P*-value | Standardized difference |
|---|---|---|---|---|
| Age (mean ± SD) | 70.5 ± 14.2 | 71.8 ± 15.2 | <0.001 | 0.58629 |
| Place of arrest | | | <0.001 | 0.05125 |
| Home residence | 19824 (69.4) | 19142 (67) | | |
| Outside of home | 8753 (30.6) | 9435 (33) | | |
| Witnessed arrest | 14632 (51.2) | 14547 (50.9) | 0.477 | −0.00595 |
| Bystander CPR | 4099 (14.3) | 4061 (14.2) | 0.65 | −0.00381 |
| Initial shockable rhythm | 1580 (5.5) | 1699 (5.9) | 0.032 | 0.01791 |
| Prehospital defibrillation | 2584 (9) | 2693 (9.4) | 0.115 | 0.01317 |
| ROSC at scene or ED | 1189 (4.2) | 1276 (4.5) | 0.073 | |
| Survival-to-admission | 5782 (20.2) | 5951 (20.8) | 0.08 | |
| Survival-to-discharge | 1553 (5.4) | 1540 (5.4) | 0.810 | |
| CPC 1 or 2 | 688 (2.3) | 713 (2.5) | 0.22 | |

SD, standard deviation; CPR, cardiopulmonary resuscitation; ROSC, return of spontaneous circulation; ED, emergency department; CPC, cerebral performance category

women were more likely to have ROSC at the scene or in the ED and had a better rate of survival to admission and discharge. No significant difference in good post-arrest CPC scale score was found between men and women. In patients aged 55 years and older, women were older than men and were more likely to have a cardiac arrest at home and an initial shockable rhythm. Women were less likely to survive admission and discharge (Table 4).

Since there were differences between both the sexes in the matched data, multivariate logistic regression analysis was performed to eliminate the confounding effect. The subgroup of patients aged 18–44 years showed no differences between men and women; therefore, there

**Table 4. Characteristics of patients with out-of-hospital arrest by age group and sex after propensity score matching.**

| Variables | 18–44 years (Total number = 3,798) | | | | ≥ 55 years (Total number = 48,756) | | | |
|---|---|---|---|---|---|---|---|---|
| | Male (N = 1,899) | Female (N = 1,899) | *P*-value | Standardized difference | Male (N = 24,378) | Female (N = 24,378) | *P*-value | Standardized difference |
| Age (mean ± SD) | 35.3 ± 7.3 | 35.2 ± 7.4 | 0.68 | −0.18251 | 75.3 ± 8.6 | 76.7 ± 9.6 | <0.001 | 1.70111 |
| Place of arrest | | | | 0.01032 | | | <0.001 | 0.03472 |
| Home residence | 1,331 (70.1) | 1,322 (69.6) | 0.75 | | 16,608 (68.1) | 16,211 (66.5) | | |
| Outside of home | 568 (29.9) | 577 (30.4) | | | 7,770 (31.9) | 8,167 (33.5) | | |
| Witnessed arrest | 973 (51.2) | 980 (51.6) | 0.82 | 0.00737 | 12,443 (51) | 12,388 (50.8) | 0.618 | −0.00451 |
| Bystander CPR | 319 (16.8) | 336 (17.7) | 0.465 | 0.02369 | 3,420 (14) | 3,378 (13.9) | 0.583 | −0.00497 |
| Initial shockable rhythm | 219 (11.5) | 222 (11.7) | 0.879 | 0.00493 | 1,139 (4.7) | 1,239 (5.1) | 0.036 | 0.01904 |
| Prehospital defibrillation | 322 (17) | 324 (17.1) | 0.931 | 0.00281 | 1,924 (7.9) | 2,004 (8.2) | 0.183 | 0.01205 |
| ROSC at scene or ED | 152 (8) | 214 (11.3) | 0.001 | | 894 (3.7) | 854 (3.5) | 0.33 | |
| Survival-to-admission | 498 (26.2) | 710 (37.4) | <0.001 | | 4,611 (18.9) | 4,437 (18.2) | 0.043 | |
| Survival-to-discharge | 218 (11.5) | 275 (14.5) | 0.006 | | 1,115 (4.6) | 1,012 (4.2) | 0.022 | |
| CPC 1 or 2 | 138 (7.3) | 168 (8.8) | 0.074 | | 409 (1.7) | 391 (1.6) | 0.521 | |

SD, standard deviation; CPR, cardiopulmonary resuscitation; ROSC, return of spontaneous circulation; ED, emergency department; CPC, cerebral performance category

**Table 5. Adjusted logistic model with propensity score matching data.**

| Primary outcomes | | ≥ 18 years (Total number = 57,514) | | ≥ 55 years (Total number = 48,756) | |
|---|---|---|---|---|---|
| | Variables | Odds ratio (95% CI) | P-value | Odds ratio (95% CI) | P-value |
| ROSC at scene or ED | | | | | |
| Age (increasing 1 year) | | 0.976 (0.974–0.979) | <0.001 | 0.967 (0.962–0.972) | <0.001 |
| Outside of home (vs. home residence) | | 2.019 (1.853–2.2) | <0.001 | 2.051 (1.856–2.267) | <0.001 |
| Initial shockable rhythm (vs. non initial shockable rhythm) | | 10.093 (9.921–11.981) | <0.001 | 10.373 (9.263–11.617) | <0.001 |
| Women (vs. men) | | 1.076 (0.988–1.172) | 0.093 | 0.952 (0.862–1.052) | 0.333 |
| Survival-to-admission | | | | | |
| Age (increasing 1 year) | | 0.978 (0.976–0.979) | <0.001 | 0.967 (0.965–0.97) | <0.001 |
| Outside of home (vs. home residence) | | 1.796 (1.72–1.876) | <0.001 | 1.785 (1.701–1.873) | <0.001 |
| Initial shockable rhythm (vs. non initial shockable rhythm) | | 3.868 (3.592–4.166) | <0.001 | 3.326 (3.05–3.626) | <0.001 |
| Women (vs. men) | | 1.044 (1.001–1.089) | 0.045 | 0.976 (0.932–1.023) | 0.315 |
| Survival-to-discharge | | | | | |
| Age (increasing 1 year) | | 0.969 (0.967–0.972) | <0.001 | 0.953 (0.949–0.958) | <0.001 |
| Outside of home (vs. home residence) | | 2.312(2.142–2.497) | <0.001 | 2.367 (2.162–2.591) | <0.001 |
| Initial shockable rhythm (vs. non initial shockable rhythm) | | 8.159 (7.456–8.929) | <0.001 | 6.881 (6.154–7.693) | <0.001 |
| Women (vs. men) | | 0.988 (0.915–1.067) | 0.752 | 0.914 (0.835–1.001) | 0.052 |
| CPC 1 or 2 | | | | | |
| Age (increasing 1 year) | | 0.958 (0.955–0.961) | <0.001 | 0.930 (0.923–0.937) | <0.001 |
| Outside of home (vs. home residence) | | 2.254 (2.006–2.532) | <0.001 | 2.604 (2.243–3.024) | <0.001 |
| Initial shockable rhythm (vs. non initial shockable rhythm) | | 15.369 (13.664–17.287) | <0.001 | 13.882 (11.922–16.164) | <0.001 |
| Women (vs. men) | | 1.082 (0.964–1.215) | 0.182 | 0.963 (0.831–1.116) | 0.620 |

CI, confidence interval; ROSC, return of spontaneous circulation; ED, emergency department; CPC, cerebral performance category

was no need to perform multiple logistic regression analysis. After adjustment for other confounders (age, place of arrest, and initial shockable rhythm), women had a higher rate of survival to admission for patients over 18 years than men. For patients aged > 55 years, women showed no differences in outcomes (Table 5).

## Discussion

In this large nationwide, population-based observational study, male patients generally had better survival outcomes. The incidence of OHCA in men was higher than that in women, which is consistent with another national registry [10, 20]. This could be explained by the higher prevalence of cardiovascular disease and lifestyle risk factors in men [21]. Many studies have shown that survival outcomes differ between the sexes, although the findings are somewhat contradictory [14, 22–25].

Before PSM, survival outcome in women was worse than men in OHCA. This could be explained by the poor prognostic characteristics of OHCA in women, such as older age, higher occurrence of arrest at home, lower witnessed arrest, bystander CPR, initial shockable rhythm, and prehospital defibrillation [9, 26]. A previous study suggested that OHCA was more likely

to occur at home in female patients, who were admitted with a negative prognosis [27]. Female patients tended to have a low survival rate due to less prehospital resuscitation efforts and the social norms of a community in attempting chest compression or defibrillation in women [3]. Female patients tended to have a lower initial shockable rhythm (ventricular fibrillation or pulseless ventricular tachycardia) than male patients; moreover, the presence of ventricular fibrillation is known to show a better prognosis than that of asystole or pulseless electrical activity, according to the latest studies [27].

In the comparison of reproductive age group (18–44 years old) of present study, there were many factors that men had a better effect on survival except being slightly older than women [20]. In other words, men had more OHCA in public places than women, and received more bystander CPR by witnesses. The initial cardiac rhythm was also subjected to more defibrillation caused by the shockable rhythm. Nevertheless, there was no difference between men and women in ROSC at scene or ED and survival-to-discharge, and survival-to-admission was higher for women. Although men have better survival factors, the failure to show better results than women seems to indicate that there are physiological differences between men and women. In menopausal women (over 55 years old), the results were similar to the all age group analyses.

After PSM, the confounding effect of three Utstein variables, that excluded the variables of age, place of arrest, and initial shockable rhythm, were eliminated for patients of all age and over 55 years groups. In the subgroup of patients aged between 18 and 44 years, all Utstein variables were matched without differences. In the all age and over 55 years groups, multiple logistic regression analyses were used to eliminate confounding effects, which were the unmatched variables. Women in the all age group (≥18 years) had a better rate of survival to admission in the final adjusted logistic model with PSM data. Hubert *et al.* [23] demonstrated the same results as ours, but Ng *et al.* [20] reported different results. In the final adjusted logistic model, the subgroup of patients over 55 years old, had no difference in survival outcomes, which was consistent with the findings of a previous study [23]. In the reproductive age group (18–44 years), in which all variables, except for good post-arrest CPC scale score variable, were matched, women had better rates of ROSC at the scene or in the ED, survival-to-admission, and survival-to-discharge. Previous studies have also shown similar results to ours [20, 23].

Physiological differences between male and female sex hormones have already been described to affect the survival rates in patients with OHCA, in the reproductive age group. Sex hormones not only have reproductive roles but are also cardioprotective and neuroprotective [28, 29]. Estrogen has a cardioprotective effect after a cardiac arrest and mediates hormonal responses in ischemia–reperfusion injuries in women of childbearing age [12, 30–32]. Although the mechanism of protection remains unclear, it appears to be related to reduced levels of lipoprotein (a) and inhibition of the oxidation of low-density lipoprotein. In animal models, estrogen was found to protect against OHCA by binding to the estrogen receptor on vascular cells and initiating the production of nitric oxide, which is required for the maintenance and repair of vascular endothelium and dilatation of vascular smooth muscle [33]. Estrogen reportedly slows down the progression of brain injury and diminishes the extent of cell death by suppressing apoptotic pathways [34]. In our study, however, no significant difference in neurological outcome of a good post-arrest CPC scale score, was observed between male and female patients. It is thought that further research will be needed on this point.

## Limitations

This study has some limitations. First, the study could not exclude uncontrolled confounders such as sex hormone levels, due to the retrospective observational and non-randomized

design. Another limitation is that there may be unmeasured confounders that could have affected the association between sex and outcome. Moreover, sex hormone levels such as those of estrogen and progesterone, were not measured in this study, and age groups were used as substitutes for hormonal levels in women.

Second, socioeconomic data were not included. This study was unable to identify the underlying socioeconomic implications of gender for OHCA results because it focused on pre-hospital factors, and socioeconomic data were not available. If differences in socioeconomic levels between men and women existed, then this would have been a confounding variable.

Third, we did not include underlying diseases such as hypertension and diabetes in patients with OHCA. Clearly, the presence or absence of an underlying disease can affect survival.

Finally, even in the PSM analysis, we are unable to exclude numerous unknown confounding factors that may mislead the sex-specific differences in outcomes after OHCA. Other limitations are common to epidemiological studies, including ascertainment bias and lack of data integrity and validity.

## Conclusion

This Korean nationwide OHCA study showed that women in the reproductive age group had better survival outcome after OHCA, when matched for confounding factors (age, location, witness and bystander presence, initial cardiac rhythm, and prehospital defibrillation). However, neurological outcomes post arrest, did not differ between men and women. Menopausal women also showed no difference in survival and neurological outcomes. On the basis of the results of this study, further studies on sex hormones are required to improve the survival rate in patients with OHCA.

## Author Contributions

**Conceptualization:** Myung Chun Kim.

**Data curation:** Han Zo Choi.

**Formal analysis:** Han Zo Choi.

**Methodology:** Han Zo Choi.

**Resources:** Han Zo Choi.

**Software:** Han Zo Choi.

**Supervision:** Han Zo Choi, Hansol Chang, Seok Hoon Ko.

**Validation:** Han Zo Choi.

**Visualization:** Han Zo Choi.

**Writing – original draft:** Han Zo Choi, Seok Hoon Ko.

**Writing – review & editing:** Han Zo Choi, Hansol Chang.

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
