## [Decision Letter · Decision Letter 0]

27 Aug 2021

PONE-D-21-23845

Gender effect in survival after out-of-hospital cardiac arrest: a nationwide, population-based, case-control propensity score matched study based Korean National Cardiac Arrest registry.

PLOS ONE

Dear Dr. MYUNG CHUN KIM,

Thank you for submitting your manuscript to PLOS ONE. After careful consideration, we feel that it has merit but does not fully meet PLOS ONE’s publication criteria as it currently stands. Therefore, we invite you to submit a revised version of the manuscript that addresses the points raised during the review process.

ACADEMIC EDITOR: Thank you very much for submitting you paper. It is potentially interesting but it has to be improved following the reviewers' comments.

We look forward to receiving your revised manuscript.

Kind regards,

Simone Savastano

Academic Editor

PLOS ONE

Journal Requirements:

2. In your ethics statement in the manuscript and in the online submission form, please provide additional information about the patient records used in your retrospective study. Specifically, please ensure that you have discussed whether all data were fully anonymized before you accessed them.

5. We noticed you have some minor occurrence of overlapping text with the following previous publication(s), which needs to be addressed:

- https://journals.lww.com/euro-emergencymed/Abstract/2021/01000/Effect_of_gender_on_out_of_hospital_cardiac_arrest.12.aspx

- https://ccforum.biomedcentral.com/articles/10.1186/s13054-019-2547-x

The text that needs to be addressed involves the Discussion and Limitations sections.

In your revision ensure you cite all your sources (including your own works), and quote or rephrase any duplicated text outside the methods section. Further consideration is dependent on these concerns being addressed."

Additional Editor Comments:

Thank you very much for submitting you paper. It is potentially interesting but it has to be improved following the reviewers' comments.

Reviewers' comments:

Reviewer's Responses to Questions

**Comments to the Author**

1. Is the manuscript technically sound, and do the data support the conclusions?

Reviewer #1: Partly

Reviewer #2: Yes

2. Has the statistical analysis been performed appropriately and rigorously? 

Reviewer #1: Yes

Reviewer #2: Yes

3. Have the authors made all data underlying the findings in their manuscript fully available?

Reviewer #1: Yes

Reviewer #2: No

4. Is the manuscript presented in an intelligible fashion and written in standard English?

Reviewer #1: No

Reviewer #2: Yes

5. Review Comments to the Author

Reviewer #1: The manuscript reported results from a propensity score matching analysis on differences in out of hospital cardiac arrest outcome according to the gender. Authors reported how in the reproductive age females have a better survival outcome than males. Meanwhile, in post-menopausal age group no differences in outcome were reported.

I have some comments for the authors:

- In the objectives of the study authors reported that they aim to analyse differences in the outcome on a gender basis, and considering subgroups of females’age. However in the abstract the results of the study were only partially reported. Particularly, results of th comparison among males and females in the post-menopausal subgroup are not reported.

- How can the authors explain the absence of significant differences in OHCA clinical presentation between males and females in reproductive age? I think that this point should be better addressed in the Discussion

- At the beginning of the discussion, authors mentioned the incidence of OHCA in their population but in the results’ session there was no reported any analysis on incidence of OHCA, thus this period should be removed from the Discussion.

- In general I found the Discussion a bit confusing, authors referred to previous studies that are in disagree with their findings but not provide a possible explanation of this disagreement

- Page 11: “Before PSM, several factors had low survival outcomes in women”. This sentence has no sense, please amend it.

Reviewer #2: Thank you for the opportunity to review your manuscript entitled "Gender effect in survival after out-of-hospital cardiac arrest: a nationwide, population-based, case-control propensity score matched study based Korean National Cardiac Arrest registry".

First of all, I would like to suggest you specify the total word count at the beginning of the manuscript and insert line numbers to make it easier to review and to correct any mistake.

Specific comments:

Abstract

-Please remove the comma between "OR" and the value. Moreover, you should write "95% CI" before the interval and at the end the p-value inside the brackets when you present your results.

Materials and Methods

-Interesting description of your EMS setting, thank you.

-Has this study an approval from the ethics committee? If yes, you should specify.

Results

-Consider rephrasing: "From January 1, 2009, to December 31, 2016 a total of 214,954 patients with OHCA were identified

from the OHCA database".

-The description of inclusion and exclusion flow chart is not very clear. Considering the numbers, it seems that no patient excluded from the study has more than one excluding criteria (i.e., under 18 years old with arrest due to trauma). Moreover, you should specify what you mean with "arrest of non-cardiac etiology", because also an arrest due to trauma has not cardiac etiology.

-Consider changing inclusion and exclusion flow chart (Fig.1): you should separate the chart about 18-44 years and >=55 years in the main flow chart you should only specify the number of female and male of the 94,160 patients included.

Discussion

-Consider changing every semicolon to punctuation in the paragraph "A previous study suggested that OHCA was more likely to occur at home in female patients (...) the presence of ventricular fibrillation is known to show a better prognosis than that of asystole or pulseless electrical activity, according to the latest studies".

-Add a punctuation at the end of the sentence "In menopausal women (over 55 years old), the results were similar to the all age group analyses".

6. PLOS authors have the option to publish the peer review history of their article (what does this mean?). If published, this will include your full peer review and any attached files.

Reviewer #1: No

Reviewer #2: No

---

## [Author Response · Author response to Decision Letter 0]

18 Sep 2021

The manuscript reported results from a propensity score matching analysis on differences in out of hospital cardiac arrest outcome according to the gender. Authors reported how in the reproductive age females have a better survival outcome than males. Meanwhile, in post-menopausal age group no differences in outcome were reported. I have some comments for the authors:

Dear reviewer.

Thank you for giving us the opportunity to submit a revised draft of the manuscript. We appreciate the effort you and you have devoted to providing your valuable feedback on the manuscript. We are thankful to you for your insightful comments on the paper. We have been able to incorporate changes to reflect the suggestions provided by you. 

Here is a point-by-point response to your comments and concerns.

1. In the objectives of the study authors reported that they aim to analyse differences in the outcome on a gender basis, and considering subgroups of females’age. However in the abstract the results of the study were only partially reported. Particularly, results of th comparison among males and females in the post-menopausal subgroup are not reported.

Thank you for your advice. We had added result of analysis of comparison among males and females in the post-menopausal subgroup in the result of abstract section. 

2. How can the authors explain the absence of significant differences in OHCA clinical presentation between males and females in reproductive age? I think that this point should be better addressed in the Discussion

We totally agree with you. We had added more discussion about this point in the third paragraph of discussion.

3. At the beginning of the discussion, authors mentioned the incidence of OHCA in their population but in the results’ session there was no reported any analysis on incidence of OHCA, thus this period should be removed from the Discussion.

The following contents were added at the beginning of the main results. 

This study found that women (N=32,345) had much less OHCA than men (N=61,915), were older than men and their OHCA occurred more at home than outside of home. They were less likely, to experience a witnessed arrest, have an initial shockable rhythm, receive bystander CPR, and prehospital defibrillation

4. In general I found the Discussion a bit confusing, authors referred to previous studies that are in disagree with their findings but not provide a possible explanation of this disagreement

In the previous study, we found that there were many opinions on the inconsistency of results. I think the discrepancy in previous studies was probably due to different analysis methods from other patient groups. Several statistical techniques were used to eliminate the limitations of the analysis presented in previous studies, such as confounding effects, as much as possible. First of all, selection bios were eliminated using the entire population data of Korea, and the PSM method was used to eliminate gender differences

5. Page 11: “Before PSM, several factors had low survival outcomes in women”. This sentence has no sense, please amend it.

Thank you for your comment. We had changed sentence as below.

Before PSM, survival outcome in women was worse than men in OHCA

Thank you for your kind comments and advice. Our manuscript has improved due to your thoughtful comments.

We look forward to hearing from you in due time regarding our submission.

Reviewer #2: Thank you for the opportunity to review your manuscript entitled "Gender effect in survival after out-of-hospital cardiac arrest: a nationwide, population-based, case-control propensity score matched study based Korean National Cardiac Arrest registry".

Dear reviewer.

Thank you for giving us the opportunity to submit a revised draft of the manuscript. We appreciate the effort you and you have devoted to providing your valuable feedback on the manuscript. We are thankful to you for your insightful comments on the paper. We have been able to incorporate changes to reflect the suggestions provided by you. 

Here is a point-by-point response to your comments and concerns.

First of all, I would like to suggest you specify the total word count at the beginning of the manuscript and insert line numbers to make it easier to review and to correct any mistake.

 Thank you

Specific comments:

Abstract

-Please remove the comma between "OR" and the value. Moreover, you should write "95% CI" before the interval and at the end the p-value inside the brackets when you present your results.

Materials and Methods

-Interesting description of your EMS setting, thank you.

 Thank you for your kind compliment. 

-Has this study an approval from the ethics committee? If yes, you should specify.

Thank you for your advice. This study was approved by the institutional review boards of the participating institutions (IRB number: 2021-05-005), and the need for informed consent was waved. We additionally mention this in front of method section as bellow 

This study was approved by the institutional review boards of the participating institutions (IRB number: 2021-05-005), and the need for informed consent was waved.

Results

-Consider rephrasing: "From January 1, 2009, to December 31, 2016 a total of 214,954 patients with OHCA were identified

from the OHCA database".

Thank you for your advice, we had changed sentence as below

“Total 214,954 patients of OHCA were identified in this study from January 1, 2009, to December 31, 2016.”

-The description of inclusion and exclusion flow chart is not very clear. Considering the numbers, it seems that no patient excluded from the study has more than one excluding criteria (i.e., under 18 years old with arrest due to trauma). Moreover, you should specify what you mean with "arrest of non-cardiac etiology", because also an arrest due to trauma has not cardiac etiology.

Thank you for your advice. We had excluded patients step by step. Therefore, there might be some patients who had more than one exclusion criteria. However, these patients had been banned in the previous step already. For example, a patient under 18 years old with arrest due to trauma had excluded in exclusion of patient in arrest due to trauma. We had added these contents in figure 1 legend as below. 

“We had excluded patient step by step from top to above. “

-Consider changing inclusion and exclusion flow chart (Fig.1): you should separate the chart about 18-44 years and >=55 years in the main flow chart you should only specify the number of female and male of the 94,160 patients included.

I totally agree with your point. We had changed figure 1 as following and clarify the separation of patients.

---

## [Decision Letter · Decision Letter 1]

4 Oct 2021

Gender effect in survival after out-of-hospital cardiac arrest: a nationwide, population-based, case-control propensity score matched study based Korean National Cardiac Arrest registry.

PONE-D-21-23845R1

Dear Dr. MYUNG CHUN KIM

We’re pleased to inform you that your manuscript has been judged scientifically suitable for publication and will be formally accepted for publication once it meets all outstanding technical requirements.

Kind regards,

Simone Savastano

Academic Editor

PLOS ONE

Additional Editor Comments (optional):

Thank you very much for having addressed all the comments of the reviewers.

Reviewers' comments:

Reviewer's Responses to Questions

**Comments to the Author**

1. If the authors have adequately addressed your comments raised in a previous round of review and you feel that this manuscript is now acceptable for publication, you may indicate that here to bypass the “Comments to the Author” section, enter your conflict of interest statement in the “Confidential to Editor” section, and submit your "Accept" recommendation.

Reviewer #1: All comments have been addressed

Reviewer #2: All comments have been addressed

2. Is the manuscript technically sound, and do the data support the conclusions?

Reviewer #1: Yes

Reviewer #2: Yes

3. Has the statistical analysis been performed appropriately and rigorously? 

Reviewer #1: Yes

Reviewer #2: Yes

4. Have the authors made all data underlying the findings in their manuscript fully available?

Reviewer #1: Yes

Reviewer #2: Yes

5. Is the manuscript presented in an intelligible fashion and written in standard English?

Reviewer #1: Yes

Reviewer #2: Yes

6. Review Comments to the Author

Reviewer #1: (No Response)

Reviewer #2: Thank you for addressing all the comments in a satisfying way, I really appreciate the changes. I have no further comments.

7. PLOS authors have the option to publish the peer review history of their article (what does this mean?). If published, this will include your full peer review and any attached files.

Reviewer #1: No

Reviewer #2: No

---

## [Editor Report · Acceptance letter]

3 May 2022

PONE-D-21-23845R1 

Gender effect in survival after out-of-hospital cardiac arrest: a nationwide, population-based, case-control propensity score matched study based Korean National Cardiac Arrest registry. 

Dear Dr. Kim:

I'm pleased to inform you that your manuscript has been deemed suitable for publication in PLOS ONE. Congratulations! Your manuscript is now with our production department. 

Kind regards, 

on behalf of

Dr. Simone Savastano 

Academic Editor

PLOS ONE